# COVID-19 Infection in Rheumatic Patients on Chronic Antimalarial Drugs: A Systematic Review and Meta-Analysis

**DOI:** 10.3390/jcm11226865

**Published:** 2022-11-21

**Authors:** Isabela Landsteiner de Sampaio Amêndola, Jonathan Aires Pinheiro, Pedro Póvoa, Vicente Cés de Souza Dantas, Rodrigo Bernardo Serafim

**Affiliations:** 1Marilia Medical School, Marília 17519-030, Brazil; 2Health Sciences Center, University of Fortaleza, Fortaleza 60811-905, Brazil; 3Siupe Primary Care Facility, São Gonçalo do Amarante 62670-000, Brazil; 4NOVA Medical School, New University of Lisbon, 1169-056 Lisboa, Portugal; 5Center for Clinical Epidemiology and Research Unit of Clinical Epidemiology, Odense University Hospital, 5000 Odense, Denmark; 6Intensive Care Department, Hospital de São Francisco Xavier, CHLO, 1150-199 Lisboa, Portugal; 7Instituto D’or de Pesquisa e Ensino, Rio de Janeiro 22281-100, Brazil; 8Clinical Research Team, Hospital Naval Marcílio Dias, Rio de Janeiro 22281-100, Brazil; 9Intensive Care Unit, Hospital Copa D’or, Rio de Janeiro 22031-011, Brazil; 10Internal Medicine Department, Universidade Federal do Rio de Janeiro, Rio de Janeiro 21941-590, Brazil

**Keywords:** coronavirus disease-19, severe acute respiratory syndrome coronavirus 2, chloroquine, hydroxychloroquine, rheumatic diseases

## Abstract

The ongoing chronic use of hydroxychloroquine or chloroquine (HCQ/CQ) in rheumatic patients might impact their outcomes after a SARS-CoV-2 infection. Therefore, we sought to assess the mortality in rheumatic patients with chronic HCQ/CQ use who developed a COVID-19 infection through a comparison between individuals chronically using HCQ/CQ with those not taking these drugs. We performed a systematic review and meta-analysis of studies on PubMed, Embase, and Cochrane Central. We included full-length reports, prospective observational cohorts, and clinical trials of adult patients (aged ≥ 18 years) who were diagnosed with a COVID-19 infection. Case studies, case series, letters, comments, and editorials were excluded. The main outcome was all-cause mortality. This study is registered with PROSPERO (CRD42022341678). We identified 541 studies, of which 20 studies were included, comprising 236,997 patients. All-cause mortality was significantly lower in patients with prior chronic use of HCQ/CQ compared to those with no previous usage (OR 0.76; 95% CI 0.62–0.94; *p* = 0.01). There was a considerably lower incidence of hospitalization among patients with chronic HCQ/CQ use compared to their counterparts without HCQ/CQ usage (OR 0.80; 95% CI 0.65–0.99; *p* = 0.04). All-cause mortality and hospitalization were significantly lower in rheumatic patients with chronic HCQ/CQ use who developed a COVID-19 infection.

## 1. Introduction

The global pandemic of coronavirus disease 2019 (COVID-19) has affected more than 633 million people and led to more than 6 million deaths across the world [1]. Some comorbidities, such as diabetes, hypertension, and advanced age, were associated with negative outcomes [2]. Patients with autoimmune rheumatic disease have underlying immune dysfunction, in addition to frequent use of diverse immunosuppressant medications. In this regard, data from the COVID-19 Global Rheumatology Alliance suggest that rheumatic patients have poorer outcomes [3].

After the outbreak of COVID-19, many medications were considered potential candidates for the treatment of a Severe Acute Respiratory Syndrome Coronavirus-2 (SARS-CoV-2) infection, including antimalarial drugs. Early in the pandemic, considerable attention was drawn to hydroxychloroquine (HCQ) and chloroquine (CQ), as in vitro studies found that these medications demonstrated antiviral activity against SARS-CoV-2 [4,5,6,7].

At the beginning of 2020, HCQ was acclaimed as a preventive and therapeutic treatment for SARS-CoV-2 infection without any scientific evidence. Correspondingly, subsequent clinical trials have not found any benefit and have shown even possible harm from HCQ use for COVID-19 infection [8,9]. In the majority of those studies, patients on the experimental arm received antimalarial medications acutely instead of chronically, which is an important element considering the pharmacokinetic (PK) and pharmacodynamic (PD) properties of these antimalarial drugs [8,9].

HCQ and CQ have very large volumes of distribution due to their extensive sequestration by tissues [10,11,12]. Consequently, a very prolonged time is needed to reach stable plasmatic concentrations, an outcome which could take as long as two months [13]. HCQ is used as an immunomodulatory drug in the treatment of systemic lupus erythematosus (SLE) and rheumatoid arthritis (RA), whereby clinical improvement is obtained only after some weeks of treatment [14]. Elevated daily doses lead to higher steady-state concentrations and a shorter time to achieve stable plasmatic concentration [13]. However, significantly high dosage is also associated with increased toxicity [14].

Therefore, we sought to perform a systematic review and meta-analysis to analyze the outcome of all-cause mortality, hospitalization, COVID-19 infection, ICU admission, need for mechanical ventilation, and requirement for oxygen therapy, drawing a comparison between rheumatic patients with chronic use of HCQ or CQ (HCQ/CQ) and those not taking these drugs.

## 2. Materials and Methods

### 2.1. Search Strategy and Selection Criteria

We performed a systematic review and meta-analysis of prospective observational studies following the recommendations of the Preferred Reporting Items for Systematic Reviews and Meta-Analyses (PRISMA) statement [15]. The protocol was registered in the Prospective Register of Systematic Reviews database (PROSPERO) under the number CRD42022341678. In order to be considered for inclusion, studies were required to meet the following criteria:Full-length reports published in peer-reviewed journals (preprint papers were excluded);Prospective observational cohorts or clinical trials of adult patients (aged ≥ 18 years);Patients were diagnosed with a COVID-19 infection through a validated test.

There were no language or time restrictions. Articles found to be case studies, case series, letters, comments, or editorials were excluded.

PubMed, Embase, and the Cochrane Library were searched. The search was conducted on 7 July 2022. We checked reference lists of retrieved articles, pertinent review articles, and personal files in order to identify relevant studies for this meta-analysis. Search terms included “hydroxychloroquine”, “chloroquine”, and “antimalarial”, which were cross-referenced with the terms “COVID-19”, “SARS-CoV-2”, and “rheumatic diseases” (see Appendix A for details of search strategy).

ILSA and VCSD screened citations identified by the initial search and selected potentially relevant titles for the review of abstracts. Among these studies, ILSA, VCSD, and RBS selected articles for the review of full-length reports. Conflicts regarding study inclusion were resolved through discussion with a third author (RBS).

Two authors (ILSA and VCSD) independently extracted data from the selected articles. When available, they recorded the following information: study characteristics (type of study, study location, type of rheumatic disease, use of glucocorticoid, number of patients enrolled, method used to diagnose COVID-19); characteristics of patients (age, sex, premorbid status); and outcomes (COVID-19 infection, all-cause mortality, hospitalization, intensive care unit (ICU) admission, need for mechanical ventilation, and requirement for oxygen therapy).

Accuracy and reliability were verified by a third author (RBS) through the extraction of data by sampling 10% of the references selected at each stage of the systematic search and, subsequently, evaluating extracted data against the original reference. Discrepancies were resolved by discussion among the authors (RBS, ILSA, VCSD, JAP, and PP). If data had not been reported, we would have contacted first or senior authors by email, although it was unnecessary as data points were readily available.

The Newcastle–Ottawa Scale was used to assess the methodological quality of included studies. This scale has been validated for the assessment of observational studies in systematic reviews and meta-analyses [16]. The scale, in which a total score ranges from 0 to 9, evaluates three aspects of study methods: study group selection (range 0–4), group comparability (range 0–2), and the quality of outcome ascertainment (range 0–3). An acceptable methodological design is reflected by a score of >5 [16].

### 2.2. Data Analysis

Characteristics of patients and outcomes (COVID-19 infection, all-cause mortality, hospitalization, ICU admission, need for mechanical ventilation, and requirement for oxygen therapy) were compared between those with and without chronic use of HCQ/CQ. Even though some studies did not report the exact time frame of treatment, in all included studies, patients on HCQ/CQ were taking these drugs to treat rheumatic diseases. In this regard, we could assume that these patients were already on HCQ/CQ and did not initiate these drugs acutely in the event of a COVID-19 diagnosis. That is why we used the word “chronic” in the paper instead of the exact period of treatment. The duration of HCQ/CQ use was reported in a few studies (Appendix B). The primary outcome of interest was all-cause mortality. The strength of the relationship between chronic HCQ/CQ use and all-cause mortality was expressed through an odds ratio (OR) with a confidence interval (CI) of 95%. Studies with zero events were entered into the analysis to include all data and reduce bias [17]. To handle studies that reported zero outcomes for mortality, we performed a series of sensitivity analyses comparing Peto, Mantel–Haenszel, and inverse variance statistical methods with fixed and random effects with 0.5 continuity correction [18]. Inverse variance and Mantel–Haenszel methods yielded identical results. We assessed publication bias by inspecting funnel plots and using the modified Egger test for binary data [19].

Heterogeneity was assessed through the I^2^ statistic, which reflects the amount of heterogeneity between studies and is robust in terms of the number of studies and the choice of effect measure [20]. To explore heterogeneity between studies, the effect of study-specific characteristics on outcome variables was estimated using meta-regression with the following predictors: age and proportion of men. The values of predictors were averaged across the groups with and without chronic HCQ/CQ use. Initially, the random-effects model was selected, on account of the potentially high heterogeneity of the included studies. If the I^2^ was <50%, a fixed-effects model was also performed. The outcome variable was the OR of all-cause mortality, with age and sex as predictors. A sensitivity analysis was executed in order to investigate the contribution of each study and to evaluate the robustness of the findings [21]. Analyses were performed utilizing RevMan version 5.4.1.

## 3. Results

### 3.1. Search Results

Figure 1 illustrates that 541 studies were identified, of which 427 did not meet the inclusion criteria and were excluded based on title or abstract review. A total of 63 studies were fully screened for inclusion. After the evaluation of exclusion criteria, 20 manuscripts remained and were included in the meta-analysis [22,23,24,25,26,27,28,29,30,31,32,33,34,35,36,37,38,39,40,41].

### 3.2. Characteristics of Included Studies

The mean age of the patients from the included studies ranged from 44 to 67 years; 34.7% were male; 82% had a diagnosis of rheumatoid arthritis; and 16% had systemic lupus erythematosus. Individual study characteristics are reported in Table 1. A total of 236,997 patients were included, of which 45,111 (19%) engaged in chronic HCQ/CQ use.

### 3.3. Risk of Bias

Based on the Newcastle–Ottawa Scale (NOS), the quality assessment of each research study included can be seen in Appendix C; studies showed an acceptable methodological design, with no study showing a high risk of bias. A sensitivity analysis was performed for all the studies included and is presented in Appendix D. The funnel plots and Egger tests are presented in Appendix E, showing a low publication bias. The meta-regression performed shows a non-significant source of heterogeneity (Appendix F).

### 3.4. Outcomes

#### 3.4.1. All-Cause Mortality

The primary endpoint of all-cause mortality after a COVID-19 infection was significantly lower in patients with previous chronic use of HCQ/CQ (105/31,289; 0.34%) compared to those with no prior use (586/165,582; 0.35%); (9 studies; OR 0.76; 95% CI 0.62–0.94; *p* = 0.01; I^2^ = 0%; Figure 2).

#### 3.4.2. Hospitalization

There was a considerably lower incidence of hospitalization due to COVID-19 infection in patients with chronic HCQ/CQ use (186/767; 24.3%) compared to the other group without chronic use of HCQ/CQ (447/1581; 28.3%); (12 studies; OR 0.80; 95% CI 0.65–0.99; *p* = 0.04; I^2^ = 0%; Figure 3).

#### 3.4.3. COVID-19 Infection

Among studies that reported the rates of COVID-19 infection, no significant difference was found in the rates of infection between chronic HCQ/CQ (192/13,691; 1.4%) versus non-chronic HCQ/CQ usage (297/26,108; 1.14%); (11 studies; OR 0.92; 95% CI 0.76–1.11; *p* = 0.38; I^2^ = 0%; Figure 4).

#### 3.4.4. ICU Admission

The rate of ICU admission due to a COVID-19 infection was not statistically different between the chronic HCQ/CQ group (8/83; 9.64%) and the non-chronic HCQ/CQ group (11/170; 6.47%); (4 studies; OR 1.55; 95% CI 0.57–4.24; *p* = 0.39; I^2^ = 0%; Figure 5).

#### 3.4.5. Mechanical Ventilation

Similarly, the need for mechanical ventilation showed no statistical difference among patients with chronic HCQ/CQ use (4/43; 9.3%) compared to individuals with non-chronic HCQ/CQ usage (10/92; 10.9%); (3 studies; OR 1.01; 95% CI 0.30–3.44; *p* = 0.99; I^2^ = 0%; Figure 6).

#### 3.4.6. Oxygen Therapy

Likewise, there was no statistical difference in the requirement for oxygen therapy between the groups, where 7 out of 44 patients (15.9%) with chronic HCQ/CQ use required oxygen therapy versus 22 out of 84 patients (26.2%) in the non-chronic HCQ/CQ group (3 studies; OR 0.54; 95% CI 0.18–1.59; *p* = 0.26; I^2^ = 0%; Figure 7).

The summary measures across the studies using fixed-effects models are provided in Appendix G.

## 4. Discussion

In this systematic review and meta-analysis, outcomes were compared between the group with prior chronic HCQ/CQ use and the group which had no previous chronic use of the aforementioned drugs, in the context of COVID-19. The main findings from the pooled population analysis were as follows: (1) the primary end-point of all-cause mortality was considerably lower in patients with prior chronic HCQ/CQ use; (2) the incidence of hospitalization due to COVID-19 infection was also notably lower in the HCQ/CQ group; (3) there was no significant difference between groups regarding rates of COVID-19 infection, ICU admission, need for mechanical ventilation, or the necessity of oxygen therapy.

HCQ/CQ have demonstrated in vitro antiviral activity against SARS-CoV-2 [4,5,6,7]. It is suggested that the antiviral effect is caused by a series of steps, which includes inhibition of the entry phase by SARS-CoV-2, as well as post-entry phases of the virus [4,5,6,7]. The drug prevents endosomal maturation and the fusion of viral and endo-lysosomal membranes by increasing endosomal pH [42]. In addition, it is believed that the immunomodulatory effect of HCQ and CQ could control the cytokine storm caused by SARS-CoV-2 through a reduction in CD154 expression by T cells [6,7]. Because of this in vitro antiviral effect of HCQ/CQ, studies were conducted to evaluate if it was possible to translate this in vitro finding to an in vivo setting.

Initially, preliminary studies on HCQ and CQ indicated a beneficial effect in treating SARS-CoV-2, as it was associated with a reduction in the time to clinical recovery, along with an improvement in COVID-19 pneumonia as shown through amelioration in chest imaging findings [43,44].

Subsequently, two extensive cohort studies, including hospitalized patients with moderate to severe COVID-19, demonstrated that HCQ—alone or in association with azithromycin—was neither associated with significant differences in the need for intubation nor in-hospital mortality [45,46].

The RECOVERY trial was a large open-label randomized trial that involved 12,000 hospitalized patients with COVID-19 in order to evaluate the efficacy of a wide variety of drugs, including HCQ, in reducing all-cause mortality within 28 days. On 5 June 2020, the RECOVERY trial reported the closure of the HCQ arm due to the absence of benefits [47].

Therefore, on 15 June 2020, the FDA revoked the authorization for the emergency use of HCQ and CQ to treat COVID-19 [48]. Subsequently, on 4 July 2020, the World Health Organization accepted the recommendation from the Solidarity Trial’s International Steering Committee to discontinue the HCQ treatment arm for hospitalized patients with COVID-19 [49].

Given those studies, there was no effect of the acute use of HCQ/CQ on COVID-19 in several outcomes, namely mortality.

It is important to reflect on the disparity between the strong in vitro antiviral effect of HCQ/CQ and the absolute lack of effect of this therapy in vivo, especially given these drugs’ PK and PD properties. The absorption of HCQ is extremely variable, ranging from 25 to 100% [50]. From a PK perspective, CQ and HCQ display a large volume of distribution (Vd) that can exceed 44,000 L in a patient with an average weight of 63.5 kg [42].

This high Vd is explained by the significant sequestration of HCQ/CQ by tissues which is specially related to the lysosomotropic characteristic of these drugs [10,11,12,42]. A considerably elevated amount of HCQ/CQ is taken into lysosomes, with an expected lysosomal accumulation ratio compared with cytosol as high as 56,000-fold [42]. The aforementioned high Vd with trapping inside endosomes explains the potential efficacy of HCQ in the case of viral pneumonia [50]. As a result of this significant Vd, these drugs present a prolonged period to achieve a steady state [13]. With a daily dosage of 0.6 mg/kg, CQ requires at least two months to achieve a steady state [13]. Accordingly, this slow accumulation of HCQ suggests that the administration of this drug as early as possible could increase the efficacy of the treatment of viral pneumonia [50].

Therefore, the prolonged period required for HCQ/CQ to achieve stable plasmatic concentration might serve to explain why chronic use of HCQ/CQ by rheumatic patients was associated with positive outcomes, whereas previous studies analyzing acute use of HCQ/CQ showed no efficacy as a treatment for COVID-19.

HCQ/CQ are diprotic weak bases dependent on the pH for lysosomal uptake and lung accumulation [51]. In viral infections, the recruitment of neutrophils associated with leaky endothelial cells results in low interstitial pH [51,52]. Accordingly, the acidic pH encountered reduces the lipid-soluble form of HCQ/CQ, which thereby decreases the uptake of these drugs by lung tissue [50]. For every pH unit of external acidification, there is a 100-fold decrease in the cellular uptake of CQ [51]. This situation worsens further in the case of a severe SARS-CoV-2 infection, where mechanical ventilation also contributes to the acidification of lung tissue [51]. Consequently, the benefits of lung sequestration of HCQ/CQ can be lost in the case of a severe COVID-19 infection [51].

Regarding PD, the full effects of HCQ or CQ therapy can take from three to six months to develop [53]. Scherbel’s group reported that, in a large open study of 805 rheumatic patients, the onset of the HCQ’s clinical effect was delayed for at least six weeks [54]. Concerning CQ, an early clinical response was also not evident in patients with rheumatic diseases since it is rare to encounter a favorable effect in less than two weeks of medication usage [55].

Our study suggests that prior chronic use of HCQ/CQ is associated with significantly lower all-cause mortality and hospitalization in rheumatologic patients who developed a COVID-19 infection compared to their counterparts with non-chronic HCQ/CQ usage. These findings could be related to the PK and PD properties of HCQ/CQ, given that plasmatic and tissue concentrations are markedly dependent on the length of time the drug was taken (acute versus chronic use).

It is also relevant to point out that malaria-endemic countries showed a low prevalence of COVID-19 coupled with a low fatality rate [56]. Although it is not possible to exclude the underestimation of COVID-19 cases, the data show a disproportionately low spread of COVID-19 in malaria-endemic regions [57]. Moreover, in African countries, no official report has documented an increase in the death rate from pneumonia of unknown causes [56]. In malaria-endemic regions, antimalarial drugs are widely used as a treatment and prevention [58]. Therefore, a strong hypothesis has already been postulated as to whether the widespread use of antimalarial drugs might have contributed to the lower mortality rate in these countries [56].

This meta-analysis has some limitations. Firstly, most studies described the use of HCQ/CQ in association with other immunosuppressors in the treatment of different rheumatic diseases. This factor contributes to the heterogeneity of the studies, limiting our ability to analyze the isolated effect of HCQ/CQ and meta-analyze data on specific subgroups. Accordingly, as it was not possible to separate patients taking other immunosuppressive drugs from those not on these medications, the use of immunosuppressants could represent a potential confounding bias in this study. Secondly, the HCQ/CQ sample sizes varied significantly, ranging from 9 patients in the smallest cohort study to 30,569 patients using HCQ/CQ in the largest sample. However, the quality of each study included was assessed in addition to a sensitivity analysis being performed to explore the effect of each study in this meta-analysis. Thirdly, some overlap of reported cases from the same database may generate some bias, although we believe that the large sample size and similarity of patients can minimize this effect. Furthermore, studies did not evaluate the degree of progression of pulmonary comorbidities, such as collagen-related lung diseases, which could affect the severity and prognosis of a COVID-19 infection. Additionally, vaccination status was not reported or evaluated in any of the included studies, which could affect COVID-19 outcomes and lead to bias. Moreover, the rheumatic diseases’ state of control was only mentioned in a few studies; therefore, a subanalysis regarding the outcomes of COVID-19 was not feasible for those groups of patients. Lastly, no study assessed long-term outcomes, which could be considered in future studies.

Regarding the strengths, this meta-analysis has resulted in an I2 statistical value of zero in all outcome analyses, showing a markedly low heterogeneity. In addition, the sensitivity analysis performed on our primary outcome (all-cause mortality) did not show any noteworthy difference when deleting any study. Furthermore, a considerable sample size of more than 236,000 patients was included in this study. Ultimately, to our best knowledge, this is the first meta-analysis evaluating the effects of chronic HCQ/CQ use by rheumatic patients on the clinical outcomes of a COVID-19 infection.

## 5. Conclusions

This meta-analysis, involving more than 236,000 patients, suggests that prior chronic use of HCQ/CQ is associated with significantly lower all-cause mortality and hospitalization in rheumatic patients who developed a COVID-19 infection compared to their counterparts with non-chronic HCQ/CQ usage.

These findings could be related to the pharmacokinetic and pharmacodynamic properties of HCQ/CQ, given that plasmatic and tissue concentrations are influenced according to the length of time the drug was taken (acute versus chronic use).

## Figures and Tables

**Figure 1 jcm-11-06865-f001:**
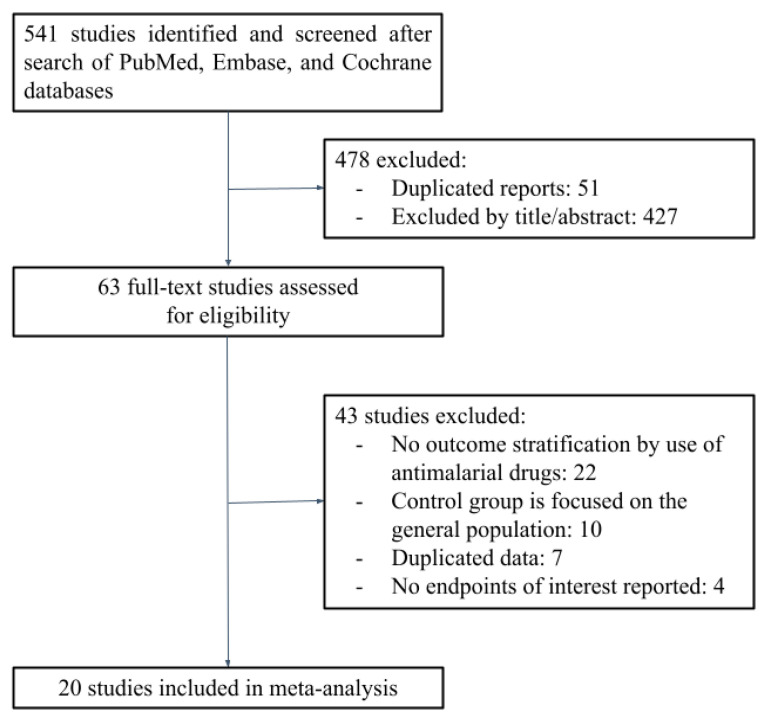
PRISMA flow diagram of study screening and selection.

**Figure 2 jcm-11-06865-f002:**
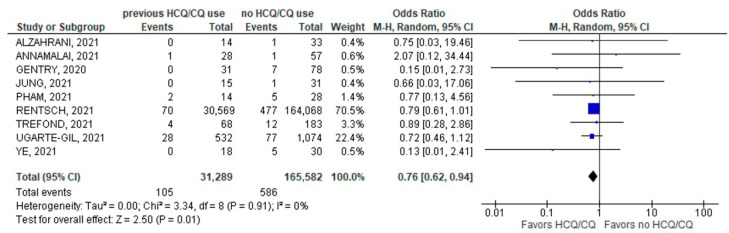
All-cause mortality [22,23,28,30,34,36,37,38,40].

**Figure 3 jcm-11-06865-f003:**
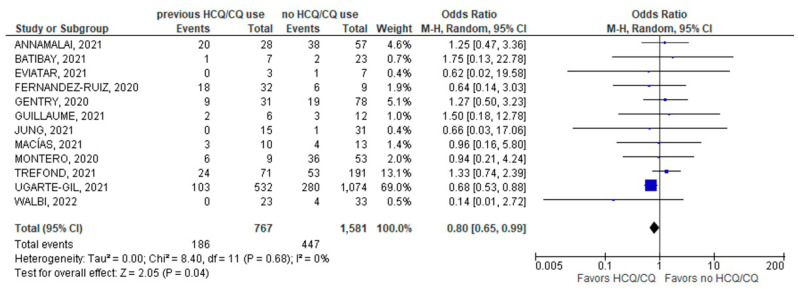
Hospitalization [23,24,26,27,28,29,30,32,33,37,38,39].

**Figure 4 jcm-11-06865-f004:**
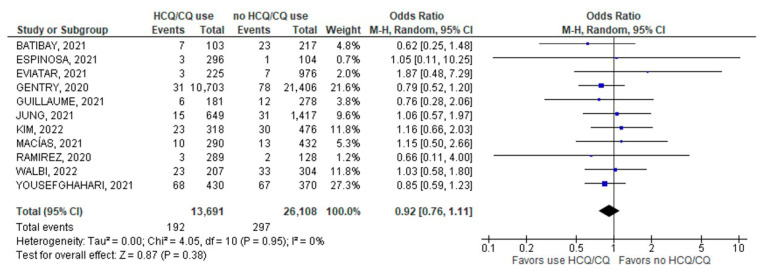
COVID-19 infection [24,25,26,28,29,30,31,32,35,39,41].

**Figure 5 jcm-11-06865-f005:**
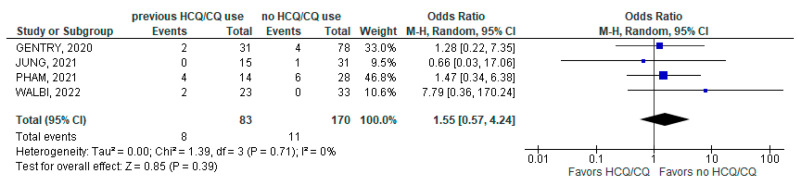
ICU admission [28,30,34,39].

**Figure 6 jcm-11-06865-f006:**
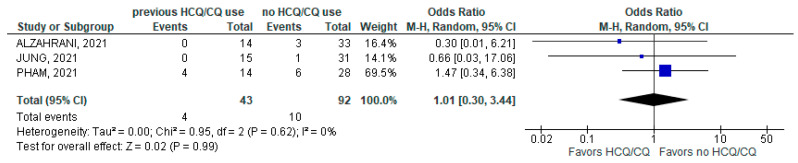
Mechanical ventilation [22,30,34].

**Figure 7 jcm-11-06865-f007:**
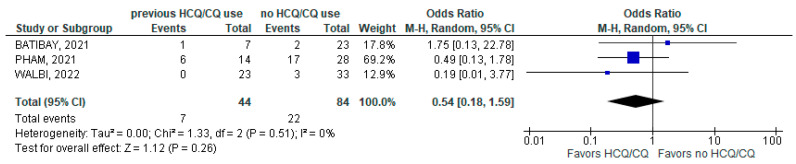
Oxygen therapy [24,34,39].

**Table 1 jcm-11-06865-t001:** Individual Study Characteristics.

Study	HCQ/CQ Users/HCQ/CQ Nonusers	HCQ/CQ/Total (%)	Male, n (%)	Mean Age, (years)	RA, n (%)	SLE, n (%)	Others, n (%)	DM, n (%)	HTN, n (%)	Lung Disease, n (%)	CVD, n (%)	Renal Disease, n (%)	CD Use, n (%)
Alzahrani, 2021 [22]	14/33	29.8	6 (12.7)	50.8	25 (53.0)	10 (21.3)	12 (25.5)	7 (14.8)	NA	NA	NA	NA	21 (44.7)
Annamalai, 2021 [23]	28/57	32.9	23 (27.0)	47.7 *	44 (51.7)	13 (15.3)	28 (32.9)	21 (24.7)	22 (25.9)	NA	NA	3 (3.5)	37 (43.5)
Batibay, 2021 [24]	103/217	32.2	91 (28.4)	47.9	109 (34.0)	17 (5.3)	194 (60.0)	35 (10.9)	70 (21.9)	36 (11.2)	28 (8.7)	13 (4.0)	113 (35.3)
Espinosa, 2021 [25]	296/104	74.0	28 (7.0)	50.7	0 (0.0)	400 (100.0)	0 (0.0)	NA	NA	NA	NA	NA	NA
Eviatar, 2021 [26]	225/976	18.7	308 (25.6)	56.0	183 (15.2)	140 (11.6)	878 (73.1)	147 (12.2)	296 (24.6)	137 (11.4)	103 (8.6)	7 (0.58)	157 (13.1)
Fernandez-Ruiz, 2020 [27]	32/9	78.0	3 (7.3)	47.0 *	0 (0.0)	41 (100.0)	0 (0.0)	NA	NA	NA	NA	NA	18 (43.9)
Gentry, 2020 [28]	10,703/ 21,406	33.3	24,531 (76.4)	65.2 *	22,242 (69.3)	7117 (22.2)	2750 (8.6)	NA	NA	6771 (21.1)	13,297 (41.4)	7632 (23.8)	2708 (8.4)
Guillaume, 2021 [29]	181/278	39.4	43 (9.3)	59.4 *	149 (32.4)	193 (42.0)	117 (25.4)	33 (7.1)	NA	33 (7.2)	167 (36.4)	NA	268 (58.4)
Jung, 2021 [30]	649/1417	31.4	574 (27.7)	62.0	1877 (90.8)	299 (14.5)	0 (0.0)	634 (30.7)	1089 (52.7)	1051 (50.9)	NA	233 (11.3)	1891 (91.5)
Kim, 2022 [31]	318/476	40.0	48 (6.0)	NA	511 (64.3)	283 (35.6)	0 (0.0)	28 (3.5)	23 (2.9)	11 (1.4)	27 (3.4)	5 (0.6)	528 (66.5)
Macías, 2021 [32]	290/432	40.2	124 (17.2)	56.5	467 (64.7)	94 (13.0)	158 (21.9)	NA	NA	NA	NA	NA	261 (36.1)
Montero, 2020 [33]	9/53	14.5	26 (41.9)	60.9	20 (32.2)	9 (14.5)	13 (21)	12 (19.3)	27 (43.5)	14 (22.6)	31 (50.0)	NA	30 (48.4)
Pham, 2021 [34]	14/28	33.3	11 (26.2)	61.0 *	18 (42.8)	7 (16.7)	19 (45.2)	12 (28.5)	29 (67.4)	16 (38.1)	NA	5 (11.9)	14 (33.3)
Ramirez, 2020 [35]	289/128	69.3	33 (7.7)	NA	0 (0.0)	417 (100.0)	0 (0.0)	12 (2.9)	123 (29.5)	40 (9.6)	NA	NA	NA
Rentsch, 2021 [36]	30,569/164,068	15.7	56,197 (28.9)	67.0 *	167,874 (86.2)	26,763 (13.7)	0 (0.0)	34,807 (17.9)	83,404 (42.8)	26,680 (13.7)	NA	26,472 (13.6)	33,677 (17.3)
Trefond, 2021 [37]	71/191	27.1	16 (6.1)	54.4 *	131 (50.0)	42 (16.0)	89 (34.0)	24 (9.2)	75 (28.6)	42 (16.3)	NA	19 (7.2)	92 (35.1)
Ugarte-Gil, 2021 [38]	665/ 1257	34.6	188 (9.8)	44.4	0 (0.0)	1922 (100.0)	0 (0.0)	NA	NA	NA	NA	223 (11.6)	825 (43.0)
Walbi, 2022 [39]	207/304	40.5	92 (18.0)	44.5	325 (63.6)	151 (29.5)	35 (6.8)	78 (15.3)	105 (20.5)	44 (8.6)	24 (4.7)	NA	138 (27.0)
Ye, 2021 [40]	18/82	18.0	60 (60.0)	59.2 *	0 (0.0)	0 (0.0)	100 (100.0)	6 (6.0)	23 (23.0)	15 (15.0)	NA	NA	22 (22.0)
Yousefghahari, 2021 [41]	430/370	53.7	NA	NA	473 (59.1)	110 (13.8)	217 (27.1)	94 (11.8)	56 (7.0)	58 (7.3)	30 (3.8)	NA	716 (89.4)

* Values calculated by this meta-analysis’s authors. HCQ/CQ: chronic hydroxychloroquine/chloroquine; RA: rheumatoid arthritis; SLE: systemic lupus erythematosus; DM: diabetes mellitus; HTN: hypertension: RPD: respiratory disease; CVD: cardiovascular disease; RND: renal disease; CD: corticosteroid.

## Data Availability

Data collected for this study will be accessible in their entirety following the publication of this article. Requests for data access can be made to the corresponding author.

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
