# Peer review of "COVID-19 Infection in Rheumatic Patients on Chronic Antimalarial Drugs: A Systematic Review and Meta-Analysis"

_jcm, 2022, doi:10.3390/jcm11226865_

Round 1

Reviewer 1 Report

The research dealt with an important topic that explains the weak efficacy of malaria drugs when acute use in late cases of Covid 19. While chronic use may benefit in reducing the risk of infection, this has been linked to good pharmacokinetics. The research represents an important scientific addition at the clinical and research level

Author Response

Point 1: The first thing is to clarify the extent to which these results of research benefit from the use of malaria medicines to prevent the Coronavirus in low-income countries, and whether it is possible to explain the lack of infections and deaths in African countries where malaria is spread and these medicines are used or are available in a general way that allows them to be prescribed and dispensed on a large scale. 

Response 1:  Regarding the relationship between the results and the prevention of coronavirus infection in low-income countries, it would be not possible to elaborate on that since the outcome of the COVID-19 infection had no statistical significance. In reference to the lack of infections and deaths in African countries, we have added a paragraph about this topic in the discussion.

Point 2: Some depth into the pharmacokinetics of malaria drugs, especially their ability to accumulate in lung tissues, as detailed below. 

Response 2: We greatly appreciate this comment as we have added a lot of information in the discussion that was elaborated based on the references provided. Therefore, we detailed the pharmacokinetics of malaria drugs by adding the following details: oral absorption; lysosomotropic characteristic of HCQ and CQ; a large volume of distribution; the relation between pH and cellular CQ uptake; association between lung acidosis and antiviral effect of HCQ and CQ.

Point 3: Uncategorized References

Response 3: We have added some uncategorized references that were used to the discussion.

Reviewer 2 Report

I would like to thank the respected editor of the Journal of Clinical Medicine for providing me the opportunity to review the manuscript entitled “Covid-19 infection in rheumatic patients on chronic antimalarial drugs: A systematic review and meta-analysis”. The authors have systematically reviewed the literature and conducted meta-analyses to compare the outcomes of COVID-19 patients with and without chronic use of antimalarial drugs. The meta-analysis results revealed that patients with chronic antimalarial drug use had a lower risk of mortality and hospitalization after COVID-19 than non-users. The manuscript is well-written and conducted according to a pre-specified protocol. The search strategy is comprehensive, and the methods are described with great attention to detail. The topic is interesting and, in my opinion, falls within the scope and readership of the Journal of Clinical Medicine. However, several points require to be addressed:

  1. The Newcastle-Ottawa scale (NOS) has 8 items; however, the comparability item can be awarded a maximum of two points. Therefore, the total NOS score ranges from 0 to 9. This point needs to be addressed in the Methods section and Table A1. In addition, in the Results section, I recommend providing a sentence indicating the results of the quality assessment of the included studies.

https://www.ohri.ca/programs/clinical_epidemiology/oxford.asp

  1. In the Results section, for ease of readers, I recommend providing the number of studies included for each of the pre-specified outcomes.

  1. In the Results section, the findings regarding publication bias (funnel plots and Egger’s test) need to be reported.

  1. In the Results section, the findings of the meta-regression should be added.

I have two further suggestions which, in my opinion, can improve the overall quality of the manuscript:

  1. If possible and reported in the included studies, I suggest providing details regarding the dose and duration of antimalarial drug administration for each of the included studies. As the authors have postulated, the chronic use of antimalarial drugs, in contrast to acute and short-term use, may result in better COVID-19 outcomes.

  1. If possible and reported in the included studies, I suggest comparing the chronic users and non-users of antimalarial drugs regarding the comorbidities described in Table 1. 

Author Response

Point 1: The Newcastle-Ottawa scale (NOS) has 8 items: however, the comparability item can be awarded a maximum of two points. Therefore, the total NOS score ranges from 0 to 9. This point needs to be addressed in the Methods section and Table A1 In addition. in the Results section, I  recommend providing a sentence indicating the results of the quality assessment of the included studies:

Response 1: We corrected the score of the NOS in the methodology as reported. Also, we added the NOS findings to the results section.

Point 2: In the Results section, for ease of readers, I recommend providing the number of studies included for each of the pre-specified outcomes.

Response 2: We have added the number of studies included for each outcome.

Point 3: In the Results section, the findings regarding publication blas. (funnel plots and Egger's test) need to be reported.

Response 3: The funnel plot and Egger's test were reported in the results, and figures were added to the appendix D.

Point 4: In the Results section the findings of the meta-regression should be added.

Response 4: The meta-regression was reported in the results, and results were described in the appendix E.

Point 5:  If possible and reported in the included studies, I suggest providing details regarding the dose and duration of antimalarial drug administration for each of the included studies. As the authors have postulated the chronic use of antimalarial drugs in contrast to acute and short term use may result in better COVID-19 outcomes.

Response 5: Unfortunately, those topics were not included in the studies. We initially thought about gathering information about the dose and duration of antimalarials, but that information was not in the studies. 

Point 6: If possible and reported in the included studies, I suggest comparing the chronic users and non-users of antimalarial drugs regarding the comorbidities described in Table 1.

Response 6: Also, we aimed to do a subanalysis by the comorbidities, however it was not possible as some studies did not bring a separate analysis regarding the diseases. 

Reviewer 3 Report

This meta-analysis study investigated the effect of chronic hydroxychloroquine (HCQ) or chloroquine (CQ) use on the prognosis of patients with COVID-19. The authors conclude that it had no effect on COVID-19 infection, ICU admission, need for mechanical ventilation, or the necessity of oxygen therapy, but was effective on all-cause mortality and incidence of hospitalization. The results of this analysis may provide new, previously unknown knowledge. However, the following improvement may be required.

Major comments

1.      The current study points out that HCQ/CQ, which is not expected to be effective when used acutely for COVID-19, may have a prognostic advantage if already used chronically. How can this be applied to the initial evaluation of the patients and subsequent treatment strategies? Please include a discussion of how the results of this study can be used as feedback for clinical practice.

2.      In rheumatoid arthritis and SLE patients, is there a bias regarding the use of immunosuppressive drugs in patients who were using HCQ/CQ and those who were not?

Minor comments

1.      Chronic respiratory disease has been reported as a risk factor for COVID-19 severity. In the present analysis, the presence or absence of collagen-related lung disease comorbid with RA or SLE could also be a factor affecting severity and prognosis. Is there any evaluation regarding these points?

2.      Please define the duration of chronic use of HCQ or CQ in this analysis. Referring to lines 61-62, would use of more than 2 months be appropriate? Please comment on this point.

3.      Please discuss the evaluation of SARS-CoV-2 vaccination history and its impact on the current analysis.

4.      It would be easier to understand if Table 1 also includes the percentage of HCQ/CQ users. Do differences in the percentage of HCQ/CQ use by study have any effect on the results? How are the outcomes for each study? Is there any difference in the outcome from study to study, as there could be differences in the prevalent strains depending on the time of analysis?

5.      Could the state of control of RA or SLE affect the prognosis of COVID-19? What is your assessment of whether there is a difference in the control of collagen disease at the onset of COVID-19 with or without the use of HCQ/CQ?

Round 2

Reviewer 3 Report

Thank you for your revision. Please provide additional responses to the followings.

Minor comments

1.       Previously, I asked about the involvement of pulmonary disease, including collagen-related lung diseases. We believe that the degree of progression of comorbid pulmonary diseases may also have an impact on the prognosis of COVID-19. Please mention if you can evaluate this point.

2.       We confirmed that the duration of HCQ/CQ administration is not clearly stated. Please consider a brief description in the text, along with the evaluation of respiratory diseases as asked in Point 1.
